# Eosinophil count trajectories are associated with the prognosis of acute myocardial infarction patients: Insights from ICU data analysis

Wen-Chao Zhang[1☯], Wen-Liang Shuai[2☯], Xiao-Qing Huang[1☯], Jin-Quan Dai[1], Jia-Hui Huo[1], Ming Shen[1], Jun-Jie Chen[1], Zhi-Ming Yang[1], Xiao-Xue Xia[3]*

1 Department of Critical Care Medicine, Changxing People's Hospital, Huzhou, China, 2 Department of Cardiovascular Medicine, The Second Affiliated Hospital, Jiangxi Medical College, Nanchang University, Jiangxi, China, 3 Department of Infectious Diseases, Changxing People's Hospital, Huzhou, China

☯ These authors contributed equally to this work.
* xiaxx2000@126.com

## Abstract

### Objective

Previous clinical studies have demonstrated conflicting evidence regarding the relationship between eosinophil (EOS) count and adverse outcomes in acute myocardial infarction (AMI). This study aimed to evaluate the impact of EOS count trajectories during ICU admission on mortality and the incidence of acute kidney injury (AKI) in AMI patients.

### Methods

A total of 1,493 critically ill AMI patients from the MIMIC-IV database were enrolled. Primary outcomes included 28-day and 1-year mortality, and secondary outcomes encompassed severe AKI incidence and ICU mortality. Group-based trajectory modeling (GBTM) was applied to identify distinct EOS count trajectories. Survival differences were assessed by Kaplan-Meier curves and log-rank tests. Associations between the EOS trajectory and mortality were evaluated using multivariable logistic/ Cox regression. Furthermore, mediation analysis was conducted to investigate the potential mediating effect of AKI on mortality.

### Results

Three EOS trajectories were identified: Trajectory1 (stable-low), Trajectory2 (low-level steady rise), and Trajectory3 (medium-level rapid rise). Compared to Trajectory1, both the Trajectory2 (HR = 0.68, 95% CI: 0.47–0.99) and Trajectory3 (HR = 0.63, 95% CI: 0.50–0.79) showed significant reductions in 28-day mortality risk. The Trajectory3 also exhibited a 34% lower 1-year mortality risk compared to Trajectory1 (HR = 0.72,

**Data availability statement:** The data analyzed in this study are from the Medical Information Mart for Intensive Care IV (MIMIC-IV, Version 3.1), a publicly accessible database. Access to the data requires registration and completion of a Data Use Agreement (DUA) via PhysioNet. Interested researchers can obtain the data at: https://physionet.org/content/mimiciv/3.1/ (DOI: https://doi.org/10.13026/kpb9-mt58). No additional raw data files are hosted by the authors because MIMIC-IV does not permit redistribution of its original data. However, all other researchers can access the same dataset using the above link and following the required procedures.

**Funding:** The project was supported by the Zhejiang Medical Association Special Fund for Clinical Medical Research - Key Project (Grant No. 2023ZYC-Z38). The funders had no role in study design, data collection and analysis, decision to publish, or preparation of the manuscript. There was no additional external funding received for this study.

**Competing interests:** The authors have declared that no competing interests exist.

**Abbreviations:** AMI, acute myocardial infarction; CABG, coronary artery bypass grafting; EOS, eosinophil; GBTM, group-based trajectory modeling; ICU, intensive care unit; KM, Kaplan-Meier; MIMIC-IV, Medical Information Mart for Intensive Care IV; RCS, restricted cubic spline.

95% CI: 0.60–0.86). Mediation analysis revealed that AKI partially mediated the association between EOS trajectories and 28-day mortality.

## Conclusion

EOS count trajectory independently predicts both short- and long-term mortality in critically ill AMI patients, establishing its role as a reliable marker for risk stratification and prognostic evaluation.

---

## Introduction

Cardiovascular diseases are the leading cause of global mortality, accounting for over 30% of total deaths worldwide and imposing a substantial socioeconomic burden [1,2]. Acute myocardial infarction (AMI), as one of the most severe manifestations of CVD, often requires intensive care unit (ICU) admission due to hemodynamic instability, and its in-hospital mortality remains persistently high [3–6]. Current risk stratification tools (e.g., SOFA and APACHE II scores) rely on static parameters measured upon admission and involve complex calculations, failing to capture dynamic clinical evolution in ICU settings. Consequently, identifying readily available, cost-effective, and clinically feasible real-time biomarkers to predict outcomes in critically ill AMI patients remains a critical unmet need in clinical research.

Eosinophils (EOS) are innate immune cells that differentiate and proliferate in the bone marrow under the regulation of growth factors including interleukin-3 (IL-3), IL-5, and granulocyte-macrophage colony-stimulating factor (GM-CSF) [7,8]. They primarily exert their effects through degranulation—releasing cytokines and enzymes—in conditions such as allergies and parasitic infections [9,10]. Emerging evidence indicates a potential association between EOS levels and cardiovascular pathophysiology [11–15]. While preclinical studies demonstrate EOS-mediated cardioprotection in experimental AMI, cardiac hypertrophy, and abdominal aortic aneurysms [16–20], substantial discrepancies persist in clinical reports regarding EOS-AMI risk correlations [21]. These paradoxical findings and limited evidence collectively underscore the critical need to elucidate the clinical significance of EOS in AMI patients.

In addition, among the myriad complications of AMI, acute kidney injury (AKI) is particularly prevalent in the critical care setting, occurring in up to 20–30% of patients and arising from a complex interplay of hemodynamic instability, contrast-induced nephropathy, systemic inflammation, and cardiorenal syndrome [22,23]. AKI independently portends worse short- and long-term outcomes following AMI, underscoring the importance of identifying early predictors of renal injury in this population. Based on this background, the present study aims to systematically evaluate the association between dynamic EOS count trajectories and all-cause mortality and acute kidney injury (AKI) in AMI patients admitted to ICU. By implementing group-based trajectory modeling (GBTM), we characterize the prognostic value of distinct EOS evolution patterns across clinical pathways, ultimately informing novel therapeutic strategies and clinical decision-making.

## Methods

### Data source

This observational study employed clinical data from the Medical Information Mart for Intensive Care-IV (MIMIC-IV v3.1), a publicly accessible critical care database comprising anonymized records of >90,000 ICU admissions at a U.S. tertiary academic medical center (2008–2022). Data access authorization was obtained after completing Collaborative Institutional Training Initiative certification (Protocol ID: 60043211), emphasizing ethical data reuse. The protocol adhered to international ethical standards, including the Declaration of Helsinki. Institutional review board approval and individual informed consent were waived owing to the retrospective design and use of fully de-identified data.

### Study population

For patients with multiple hospital admissions, only data from their first admission were analyzed. In cases where patients experienced multiple ICU transfers during their initial hospitalization, clinical records from the first ICU admission were exclusively extracted. As detailed in Fig 1, 6,864 AMI patients admitted to ICU were identified. Exclusion criteria were: 1) Age < 18 years (n = 0); 2) ICU length of stay <24 hours (n = 1,056); 3) Hematologic/rheumatic diseases, parasitic infections, or asthma/allergic conditions (n = 155); 4) Corticosteroid use (n = 373); and 5) fewer than 2 EOS counts measured within the first 7 days of ICU admission (n = 3,787). The final analytical cohort comprised 1,493 AMI patients.

### Data extraction and definitions

Data extraction and processing were performed using Structured Query Language to retrieve demographics, comorbidities, laboratory parameters, and therapeutic interventions. For baseline clinical variables (vital signs and laboratory tests),

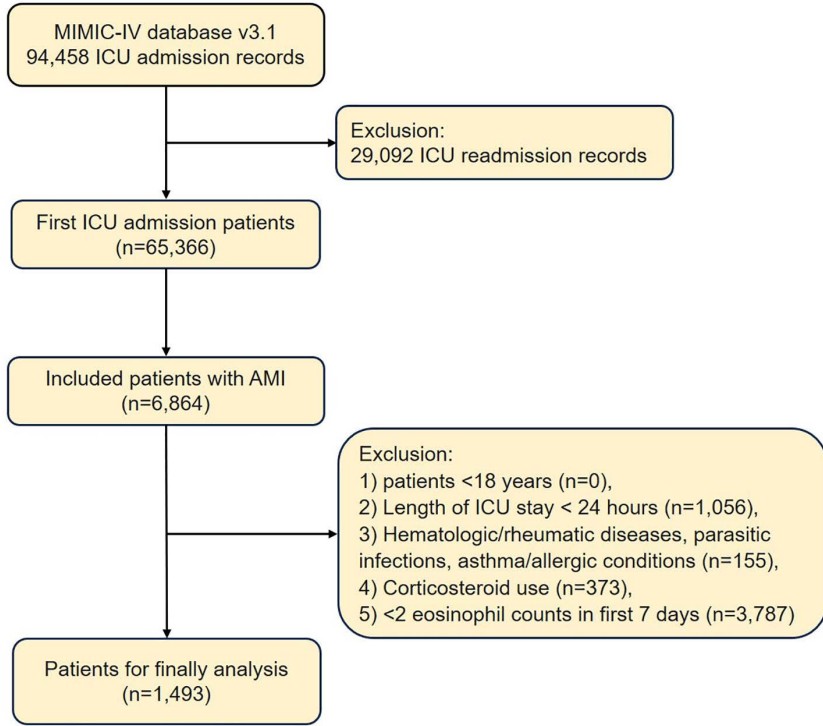

**Fig 1. Flowchart of patient inclusion and exclusion from the MIMIC-IV database.**

the first recorded value within 24 hours post-ICU admission was extracted; when multiple measurements existed within this window, the earliest value was prioritized to mitigate treatment-related confounding. EOS counts measured during the initial seven ICU days were used to compute dynamic trajectories through GBTM, which classified patients into three distinct patterns: stable-low, low-level steady rise, and medium-level rapid rise trajectories. The average posterior probabilities (AvePP) for each of the three trajectories were 0.998, 0.995, and 0.999, respectively, and the odds of correct classification (OCC) were 1579.4, 247.6, and 946.5—all well above the recommended thresholds of 0.7 and 5. Variables exceeding 15% missingness—BMI (15.61%) and cTnT (15.20%)—were imputed by random forest algorithms to preserve statistical power and minimize bias.

## Study endpoint

The primary endpoint was 28-day and 1-year all-cause mortality. The MIMIC-IV database ensures complete 1-year follow-up data for all included patients, with death information sourced from state death registries and hospital medical records. Secondary endpoints included: Severe acute kidney injury (AKI) incidence and ICU mortality. AKI was diagnosed per Kidney Disease: Improving Global Outcomes (KDIGO) criteria, defined as meeting ≥1 of: 1) serum creatinine (SCr) increase ≥1.5-fold from baseline within 7 days of admission; 2) absolute SCr rise ≥0.3 mg/dL within 48 hours; or 3) urine output <0.5 mL/kg/h for ≥6 consecutive hours [24]. Severe AKI was specifically defined as stage ≥2 KDIGO classification. Baseline SCr was determined as the lowest value within 7 days pre-hospitalization; when unavailable, the initial admission SCr measurement served as reference [25,26].

## Statistical analysis

Baseline characteristics were summarized using descriptive statistics appropriate to data distribution: continuous variables following normal distributions were expressed as mean ± standard deviation, non-normally distributed variables as median with interquartile range (IQR), and categorical variables as frequency counts with percentages. Group comparisons were conducted using Student's t-test or ANOVA for normally distributed continuous variables, Mann-Whitney U or Kruskal-Wallis test for nonparametric continuous variables, and Pearson's chi-square or Fisher's exact tests for categorical variables based on data characteristics.

GBTM was employed to characterize distinct eosinophil count evolution patterns during the initial ICU stay, assuming a normal distribution and first-order (linear) polynomial terms [27]. The optimal trajectory model was selected through an iterative process evaluating: Bayesian information criterion (BIC), Akaike information criterion (AIC), average posterior probabilities (AvePP > 0.7), odds of correct classification (OCC > 5), minimum subgroup size (>5% of cohort), and clinical interpretability.

The primary endpoints of 28-day and 1-year all-cause mortality were analyzed using Kaplan-Meier (KM) survival curves with log-rank tests and multivariable Cox regression; proportional hazards assumptions were validated by Schoenfeld residual tests. Secondary outcomes included severe AKI incidence and ICU mortality, assessed by logistic regression. Variance inflation factors (VIF < 5) confirmed no significant multicollinearity (S1 Table). All models were progressively adjusted: Model 1 (unadjusted), Model 2 (adjusted for demographics including age, gender, and BMI), and Model 3 (adjusted for covariates with univariate significance (P < 0.05, S2 Table) combined with clinically relevant variables including age, gender, BMI, SBP, Heart rate, HB, WBC, PLT, Scr, Bun, cTnT, AF, CKD, ACEI/ARB, Beta blocker, Antiplatelet drugs, Statin, PCI and CABG). Mediation analysis with bootstrap-derived confidence intervals (1,000 resamples) quantified severe AKI's indirect effect on 28-day mortality. Subgroup analyses examined interaction effects by forest plots. Sensitivity analyses were performed through restricted cubic splines (four knots) for admission and the last EOS counts alongside trajectory reclassification by EOS percentages.

Statistical significance was defined as a two-tailed *P*-value <0.05. All analyses were performed using R software (version 4.3.2).

## Results

### Baseline characteristics

This study enrolled 1,493 critically ill AMI patients with a mean age of 72 years, including 549 females (36.77%). GBTM estimated distinct posterior probabilities of trajectory group membership for each patient. Based on the aforementioned selection criteria—integrating BIC, AIC, AvePP, OCC, minimum subgroup size, and clinical interpretability—patients were classified into three distinct trajectory groups: Trajectory 1 (stable-low); Trajectory 2 (low-level steady rise); Trajectory 3 (medium-level rapid rise). The trajectory distribution is illustrated in Fig 2, with corresponding model parameters detailed in S3–S5 Tables.

Table 1 presents baseline characteristics stratified by trajectory groups. The stable-low trajectory group (Trajectory 1) demonstrated significantly older age, higher male predominance, lower BMI, faster heart rate, reduced hemoglobin levels, and elevated serum creatinine, blood urea nitrogen and APSIII compared to other trajectories. Furthermore, Trajectory 2 (low-level steady rise) and Trajectory 3 (medium-level rapid rise) groups exhibited greater utilization of guideline-directed medications and higher rates of coronary artery bypass grafting (CABG) versus the stable-low group (all P < 0.05).

### Associations of EOS counts trajectory and adverse outcomes

KM survival curves and Cox regression analyses were employed to investigate associations between eosinophil trajectories and mortality. KM analysis demonstrated significantly reduced survival probabilities in the stable-low trajectory group (Trajectory 1) at both 28 days and 1 year (log-rank P < 0.001; Fig 3). Furthermore, compared to Trajectory 1, both Trajectory 2 (low-level steady rise; HR = 0.68, 95% CI: 0.47–0.99) and Trajectory 3 (medium-level rapid rise; HR = 0.63, 95% CI: 0.50–0.79) were associated with reduced 28-day mortality. After multivariable adjustment, Trajectory 3 remained an independently associated with a lower risk of 1-year mortality (HR = 0.72, 95% CI: 0.60–0.86), whereas Trajectory 2 lost statistical significance (Table 2). Further analyses revealed that compared to Trajectory 1, both Trajectory 2 and Trajectory 3 were inversely associated with ICU mortality and severe AKI incidence during hospitalization (both *P* < 0.05; S6 Table). These findings collectively indicate that rising eosinophil counts predict improved clinical outcomes in AMI patients.

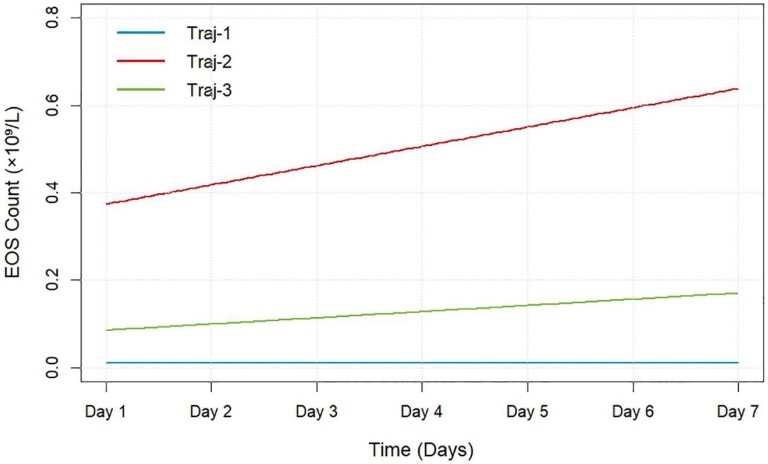

**Fig 2. EOS count trajectories over the first 7 days of ICU admission.** Trajectory1 (35.83%): stable-low group; Trajectory2 (13.40%): Low-level steady rise group; Trajectory3 (50.77%): Medium-level rapid rise group.

**Table 1. Baseline characteristics of AMI participants according to EOS count trajectories.**

| Variables | Overall (n = 1493) | Trajectory1 (35.83%) | Trajectory2 (13.40%) | Trajectory3 (50.77%) | P value |
|---|---|---|---|---|---|
| Demographics | | | | | |
| Age, years | 71.76 ± 13.46 | 73.50 ± 13.50 | 71.75 ± 13.63 | 70.53 ± 13.27 | <0.001 |
| Female, n (%) | 549 (36.77%) | 220 (41.12%) | 70 (35.00%) | 259 (34.17%) | 0.033 |
| BMI, kg/m2 | 28.69 ± 6.12 | 28.01 ± 5.89 | 29.44 ± 6.70 | 28.97 ± 6.07 | 0.004 |
| Vital signs | | | | | |
| SBP, mmHg | 121.28 ± 24.30 | 120.94 ± 24.99 | 120.54 ± 26.41 | 121.72 ± 23.23 | 0.767 |
| DBP, mmHg | 68.77 ± 19.39 | 70.02 ± 20.79 | 65.93 ± 20.38 | 68.63 ± 17.98 | 0.037 |
| HR, bpm | 90.22 ± 21.05 | 93.17 ± 22.09 | 89.11 ± 22.14 | 88.42 ± 19.76 | <0.001 |
| Comorbidities | | | | | |
| Hypertension, n (%) | 1262 (84.53%) | 443 (82.80%) | 172 (86.00%) | 647 (85.36%) | 0.378 |
| CHF, n (%) | 874 (58.54%) | 310 (57.94%) | 126 (63.00%) | 438 (57.78%) | 0.387 |
| AF, n (%) | 609 (40.79%) | 225 (42.06%) | 87 (43.50%) | 297 (39.18%) | 0.412 |
| DM, n (%) | 697 (46.68%) | 235 (43.93%) | 98 (49.00%) | 364 (48.02%) | 0.271 |
| CKD, n (%) | 496 (33.22%) | 178 (33.27%) | 86 (43.00%) | 232 (30.61%) | 0.004 |
| Laboratory measurements | | | | | |
| HB, g/dL | 11.23 ± 2.66 | 11.12 ± 2.60 | 10.75 ± 2.59 | 11.42 ± 2.71 | 0.004 |
| WBC, $10^9$/L | 12.05 (8.60-16.50) | 12.80 (8.80-17.20) | 13.05 (9.88-18.30) | 11.20 (8.20-14.90) | <0.001 |
| PLT, $10^9$/L | 223.90 ± 115.99 | 211.21 ± 117.55 | 247.24 ± 135.80 | 226.84 ± 108.15 | <0.001 |
| Scr, mg/dL | 1.40 (1.00-2.20) | 1.50 (1.00-2.30) | 1.40 (1.00-2.40) | 1.30 (0.90-2.00) | <0.001 |
| Bun, mg/dL | 28.00 (18.00-48.00) | 32.00 (20.00-51.00) | 30.00 (19.00-50.00) | 25.00 (17.00-45.00) | <0.001 |
| cTnT, ng/mL | 0.68 (0.16-1.66) | 0.65 (0.17-1.60) | 0.59 (0.16-1.50) | 0.72 (0.16-1.84) | 0.183 |
| APSIII | 53.49 ± 23.09 | 59.54 ± 24.40 | 50.45 ± 19.92 | 50.02 ± 22.05 | <0.001 |
| Treatments | | | | | |
| ACEI/ARB, n (%) | 607 (40.66%) | 164 (30.65%) | 95 (47.50%) | 348 (45.91%) | <0.001 |
| Beta blocker, n (%) | 1181 (79.10%) | 384 (71.78%) | 173 (86.50%) | 624 (82.32%) | <0.001 |
| Antiplatelet drugs, n (%) | 1250 (83.72%) | 415 (77.57%) | 174 (87.00%) | 661 (87.20%) | <0.001 |
| Statin, n (%) | 1226 (82.12%) | 394 (73.64%) | 174 (87.00%) | 658 (86.81%) | <0.001 |
| PCI, n (%) | 127 (8.51%) | 34 (6.36%) | 18 (9.00%) | 75 (9.89%) | 0.077 |
| CABG, n (%) | 255 (17.08%) | 42 (7.85%) | 47 (23.50%) | 166 (21.90%) | <0.001 |
| Outcomes | | | | | |
| ICU mortality, n (%) | 235 (15.74%) | 131 (24.49%) | 15 (7.50%) | 89 (11.74%) | <0.001 |
| Severe AKI incidence, n (%) | 1034 (69.26%) | 407 (76.07%) | 133 (66.50%) | 494 (65.17%) | <0.001 |
| 28-day mortality, n (%) | 377 (25.25%) | 203 (37.94%) | 37 (18.50%) | 137 (18.07%) | <0.001 |
| 1-year mortality, n (%) | 622 (41.66%) | 292 (54.58%) | 76 (38.00%) | 254 (33.51%) | <0.001 |

Data are mean ± SD or median (IQR) or (%).

Abbreviations: ACEI/ARB: angiotensin-converting enzyme inhibitors/angiotensin receptor blockers; AF: atrial fibrillation; AKI: acute kidney injury; AMI: acute myocardial infarct; APSIII: Acute Physiology and Chronic Health Evaluation III; BMI: body mass index; Bun: Blood urea nitrogen; CABG: coronary artery bypass grafting; CHF: congestive heart failure; CKD: chronic kidney disease; cTnT: cardiac troponin T; DBP: diastolic blood pressure; DM: diabetes mellitus; EOS: Eosinophil; HB: hemoglobin; HR: heart rate; PCI: percutaneous coronary intervention; PLT: platelet; SBP: systolic blood pressure; Scr: Serum creatinine; WBC: white blood cell.

## Mediating effect

A mediation analysis was conducted to explore the potential mediating role of AKI in the association between EOS count trajectories and 28-day mortality. Compared to trajectory 1, trajectory 2 and 3 exhibited significantly lower rates of severe

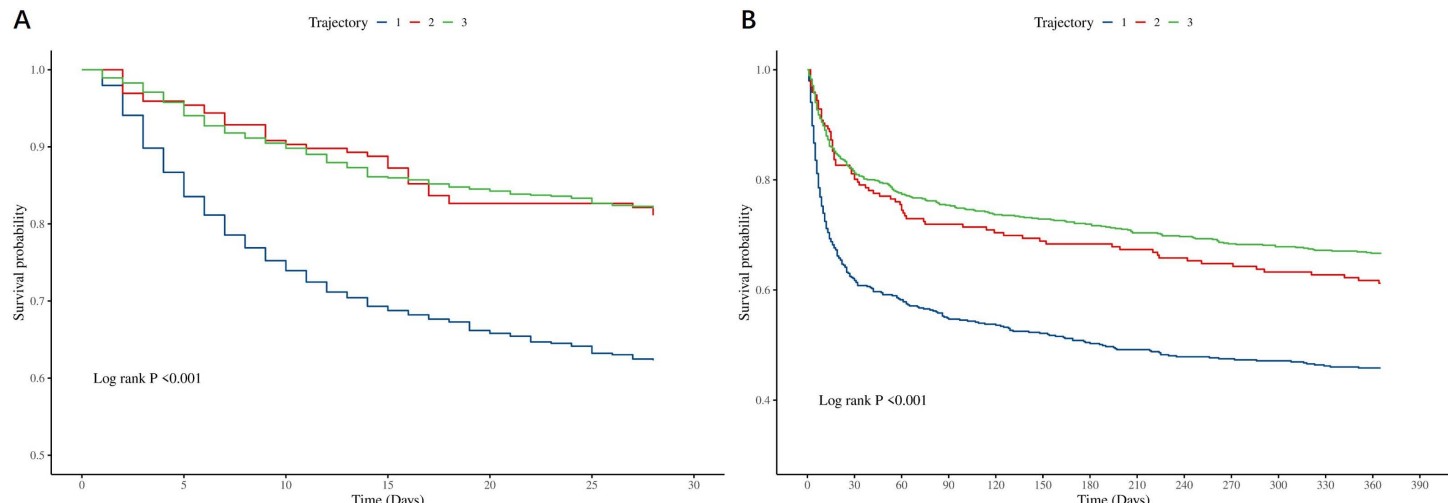

**Fig 3. Kaplan–Meier curves for (A) 28-day and (B) 1-year mortality according to EOS count trajectory.**

**Table 2. The associations of EOS count trajectories with 28-day and 1-year mortality in AMI patients.**

| | Model1 | | Model2 | | Model3 | |
|---|---|---|---|---|---|---|
| | HR (95%CI) | P value | HR (95%CI) | P value | HR (95%CI) | P value |
| **28-day mortality** | | | | | | |
| Trajectory1 | Ref | | Ref | | Ref | |
| Trajectory2 | **0.42 (0.30, 0.60)** | **<0.001** | **0.43 (0.30, 0.61)** | **<0.001** | **0.68 (0.47, 0.99)** | **0.045** |
| Trajectory3 | **0.41 (0.33, 0.51)** | **<0.001** | **0.43 (0.35, 0.54)** | **<0.001** | **0.63 (0.50, 0.79)** | **<0.001** |
| **1-year mortality** | | | | | | |
| Trajectory1 | Ref | | Ref | | Ref | |
| Trajectory2 | **0.57 (0.44, 0.73)** | **<0.001** | **0.59 (0.46, 0.76)** | **<0.001** | 0.86 (0.66, 1.13) | 0.307 |
| Trajectory3 | **0.50 (0.42, 0.59)** | **<0.001** | **0.53 (0.45, 0.63)** | **<0.001** | **0.72 (0.60, 0.86)** | **<0.001** |

Model1: unadjusted.

Model2: adjusted for age, gender, BMI.

Model3: adjusted for age, gender, BMI, SBP, HR, HB, WBC, PLT, Scr, Bun, cTnT, HF, AF, CKD, APSIII, ACEI/ARB, Beta blocker, Antiplatelet drugs, Statin, PCI, CABG.

Abbreviations as in Table 1.

AKI (KDIGO stage ≥2) during hospitalization ($P<0.001$; Tables 1 and S6). These groups were also independently associated with lower AKI risk (trajectory 2: OR = 0.62, 95% CI: 0.42–0.92; trajectory 3: OR = 0.63, 95% CI: 0.48–0.82). Bootstrap analysis indicated AKI may statistically account for part of the association between EOS trajectories and mortality. After covariate adjustment, the proportion mediated by AKI was 11.87% (95% CI: 1.39–28.02%; S7 Table). Collectively, these results suggest that severe AKI could partially explain the short-term mortality risk linked to distinct EOS count trajectory patterns.

## Subgroup analysis

Subgroup analysis results are shown in Fig 4. When stratified by demographic and clinical characteristics—including age, sex, BMI, hypertension, diabetes, and CABG—the association between EOS count trajectories

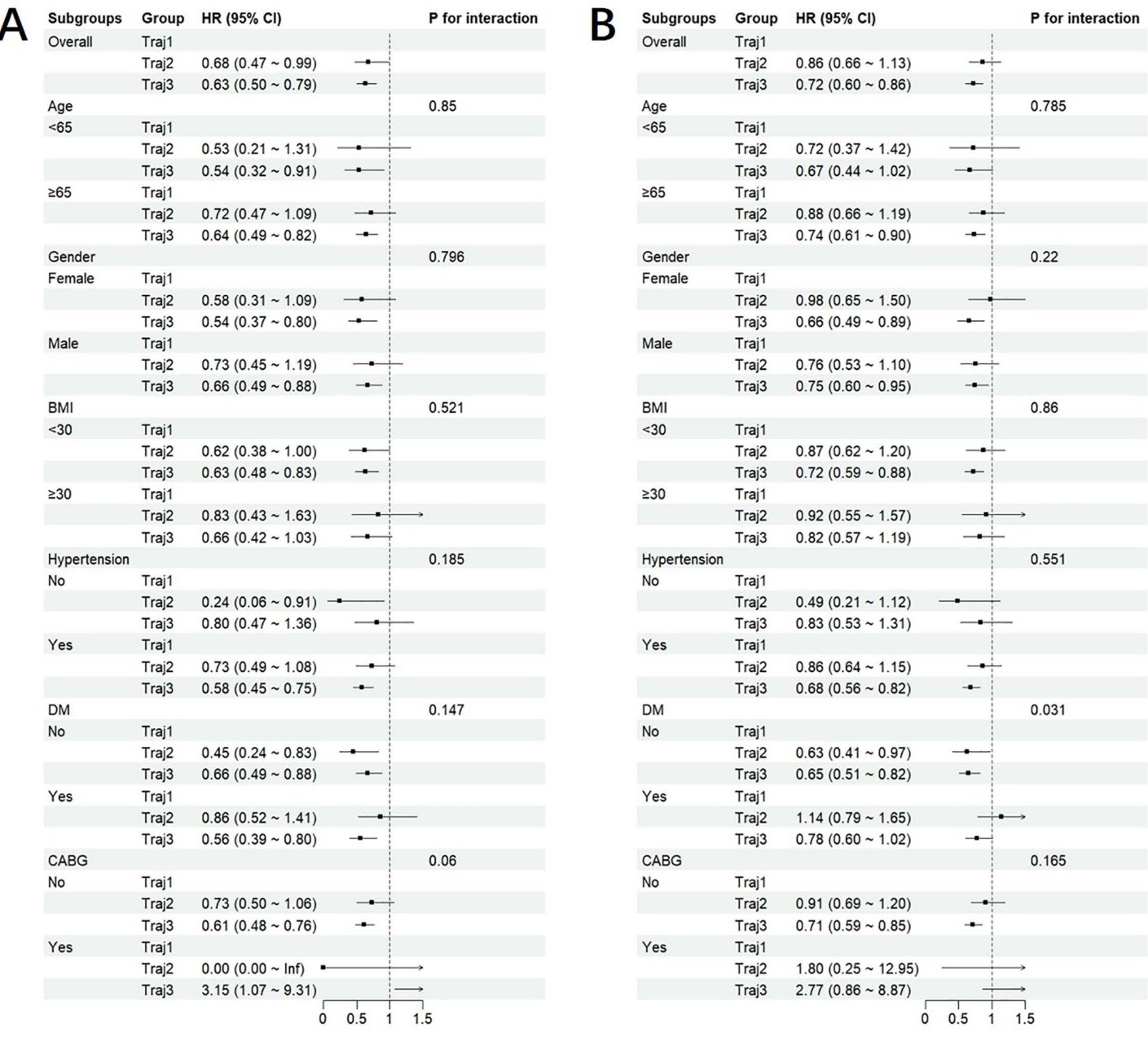

**Fig 4. Subgroup analysis for risk of in (A) 28-day and (B) 1-year mortality according to EOS count trajectory.**

and short-term mortality remained consistent across all subgroups (all interaction *P* values > 0.05). However, for long-term mortality, the inverse association between elevated EOS trajectories (trajectory 2 and 3) and mortality was significantly more pronounced in AMI patients without diabetes compared to those with diabetes (interaction *P* value = 0.031).

## Sensitivity analysis

Sensitivity analyses confirmed the robustness of the primary findings. RCS analysis (S1 Fig) revealed significant L-shaped associations of both baseline and final EOS counts with 28-day and 1-year mortality (all $P<0.001$), indicating that lower EOS levels consistently predicted adverse prognosis.

Furthermore, to explore the impact of distinct EOS manifestations on mortality, we conducted comparative analyses between trajectory patterns derived from absolute counts versus those based on relative percentages. KM analysis of EOS percentage trajectories (S2 Fig) replicated the primary trajectory findings: Trajectory 1 (stable-low) exhibited significantly reduced survival probabilities at both 28 days and 1 year compared to other trajectories (log-rank test $P<0.001$). Additionally, multivariable Cox regression analysis (S8 Table) confirmed that Trajectory 2 (low-level steady rise) and 3 (medium-level rapid rise) remained independently associated with lower risks of both short-term and long-term mortality (all $P<0.05$).

## Discussion

This cohort study systematically evaluated the prognostic significance of EOS count trajectories in critically ill AMI patients through trajectory modeling and mediation analysis. Key findings include: (1) A persistently low EOS trajectory (Trajectory 1: stable-low) independently predicted significantly increased risks of both 28-day and 1-year mortality; (2) In contrast, trajectories characterized by sustained elevation—whether gradual (Trajectory 2: low-level steady rise) or rapid (Trajectory 3: medium-level rapid rise)—were independently associated with a reduced risk of short- and long-term mortality; (3) Mechanistically, severe AKI partially mediated the association between EOS trajectories and 28-day mortality; This study demonstrates that dynamic EOS trajectories effectively predict prognosis in critically ill AMI patients, with sustained rising patterns independently associated with significantly reduced mortality risk.

Following AMI, the inflammatory system becomes activated during the hyperacute phase of plaque rupture and thrombosis, significantly preceding the activation of the sympathetic nervous system and the renin-angiotensin-aldosterone system. During the acute ischemic phase, massive infiltration of immune cells—including neutrophils, EOS, lymphocytes, and monocytes/macrophages—into necrotic myocardial tissue releases cytokines that subsequently activate innate immunity and trigger intense inflammatory responses. Certain interactions among these immune cells may facilitate early ischemic tissue healing and cardiomyocyte repair [28]. Mechanistic investigations reveal that eosinophil cationic protein protects cardiomyocytes against hypoxia- and pressure overload-induced death, attenuates cardiomyocyte hypertrophy, suppresses TGF-β-induced profibrotic protein expression in cardiac fibroblasts, inhibits angiogenesis, and suppresses inflammatory cell activation [29]. Moreover, EOS may alleviate post-MI cardiac dysfunction through secretion of IL-4 and EOS-derived ribonuclease 1 (mEar1 in mice), reducing cardiomyocyte apoptosis, fibroblast activation, and neutrophil adhesion. Both genetically EOS-deficient mice and diphtheria toxin-mediated EOS depletion models exhibit exacerbated MI pathology and heart failure following coronary ligation [18]. Further mechanistic evidence indicates that group 2 innate lymphoid cells contribute to myocardial protection by promoting EOS differentiation and maturation via IL-5 secretion—reinforcing the cardioprotective potential of EOS in MI pathogenesis [19]. These preclinical studies substantiate our clinical observations: the persistently low EOS trajectory observed early in ICU admission indicates that eosinopenia compromises endogenous cardioprotective mechanisms, thereby explaining the elevated short- and long-term mortality. Notably, the covariate-adjusted protective effect of Trajectory 2 (low-level steady rise) became non-significant for long-term mortality, suggesting that sustained higher EOS thresholds may be required to durably activate cardioprotective pathways.

However, clinical studies present divergent conclusions regarding the role of EOS in cardiovascular diseases, with most prior investigations relying on single-point EOS measurements. A study of 1,543 patients with perioperative MI or non-urgent PCI revealed elevated blood eosinophil counts in male patients and those with hypertension, prior revascularization, or receiving medical therapy [30]. Additionally, a Danish randomized controlled trial demonstrated higher EOS counts in males with prior AMI history than those without, with multivariable logistic regression identifying elevated blood EOS as a significant risk factor for

human AMI [31]. These findings suggest pathogenic roles of EOS in post-MI cardiac injury. In contrast, Gao et al. enrolled 5,287 patients undergoing coronary angiography to evaluate associations between biochemical markers (including EOS counts) and coronary stenosis severity quantified by the Gensini score—a quantitative tool where higher scores indicate more severe stenosis [32]. Their results indicated lower EOS percentages among total leukocytes in patients, showing negative correlation with Gensini scores. Moreover, acute-phase AMI patients also exhibited significantly low EOS percentages. These discrepancies may stem from differing data capture timings capturing EOS values at distinct disease phases, thereby altering prognostic associations. Notably, this study focuses on dynamically monitored EOS trajectories during ICU hospitalization—rather than single-point measurements. Our study reveals that although Trajectory 2 initially exhibited low EOS levels comparable to Trajectory 1, its subsequent gradual increase likely underlies divergent clinical outcomes—highlighting the superior prognostic value of dynamic EOS monitoring over single-point measurements in AMI patients. This study expands the potential application scope of EOS in prognostic assessment for cardiovascular critical care.

Through mediation modeling, our analysis suggests for the first time that EOS trajectory patterns may provide a partial explanation for short-term mortality risk through their statistical association with AKI. This raises critical pathophysiological questions: Within AMI's unique "cardio-renal interplay" context (e.g., contrast-induced AKI, inflammatory cascades, oxidative stress, and cardiorenal syndrome) [33–35], how does EOS-mediated immunomodulation contribute to AKI pathogenesis? Previous studies indicate higher risks associated with low EOS in patients with Thrombolysis in Myocardial Infarction (TIMI) 0 flow or elevated Gensini scores [36]. We thus postulate that persistently low EOS levels reflect severe coronary stenosis and suboptimal revascularization, ultimately causing more pronounced impairment of cardiac pump function and subsequent renal hypoperfusion. Prospective multicenter studies are warranted to elucidate the underlying mechanisms. Additionally, subgroup analysis revealed that the protective effect of elevated eosinophils on long-term outcomes was diminished in AMI patients with diabetes, suggesting that hyperglycemia-induced oxidative stress and inflammatory responses may exacerbate myocardial injury [6,37].

As a routine component of complete blood count, EOS offer high accessibility and cost-effectiveness. This study innovatively demonstrates that dynamic trajectory analysis of EOS enables early and precise risk stratification in the ICU setting. For patients with persistently low EOS trajectories, clinicians should implement intensified in-hospital monitoring—including dynamic renal function assessment with AKI-preventive interventions and hemodynamics-guided cardiac support—alongside establishing structured post-discharge follow-up protocols. This risk stratification model transforms conventional laboratory parameters into dynamic decision-making tools, providing a novel paradigm for precision management in critical cardiovascular care.

## Strengths and limitations

The strengths of this study include its innovative application of dynamic trajectory modeling to characterize temporal EOS dynamics in critically ill AMI patients—overcoming limitations of static single-point measurements—alongside multidimensional endpoint validation integrating mediation analysis that firstly identified AKI as a partial mediator of the EOS-mortality relationship.

However, several limitations warrant consideration. First, the single-center retrospective design may introduce selection bias, and our cohort represents critically ill AMI patients requiring ICU care, which limits applicability to general AMI populations. Validation of our findings in prospective multicenter cohorts is warranted. Second, despite adjustment for multiple confounders, residual bias from unmeasured variables may persist. Third, inherent database limitations precluded access to cardiac function assessments such as echocardiography. The absence of data on left ventricular ejection fraction, wall motion abnormalities, or other structural parameters represents an important constraint, as cardiac function may influence both the systemic inflammatory response and long-term survival. Finally, the 7-day observation window required for trajectory modeling limits real-time applicability during early ICU admission, and future studies should explore whether abbreviated trajectories derived from the first 48–72 hours retain comparable prognostic value. Future studies should incorporate more comprehensive datasets to robustly validate these associations.

## Conclusions

Early eosinophil trajectories in ICU patients with acute myocardial infarction were strongly associated with mortality and AKI. Persistently low eosinophil counts identified patients at highest risk, while rising trajectories predicted better outcomes. As eosinophil counts are routinely available, trajectory-based assessment may serve as a simple tool for early risk stratification, warranting prospective validation.

## Supporting information

**S1 Table. Multicollinearity Diagnostics Using Generalized Variance Inflation Factors (GVIF).**
(DOCX)

**S2 Table. Univariate Cox regression analysis for 28-day and 1-year mortality.**
(DOCX)

**S3 Table. The Group-based Trajectory Modelling (GBTM) parameters (AIC, BIC and class sizes) for EOS count trajectory grouping.** AIC: Akaike Information Criterion; BIC: Bayesian Information Criteria.
(DOCX)

**S4 Table. The Group-based Trajectory Modelling (GBTM) parameters (AvePP) for EOS count trajectory grouping.**
AvePP: Average Posterior Probabilities.
(DOCX)

**S5 Table. The Group-based Trajectory Modelling (GBTM) parameters (OCC) for EOS count trajectory grouping.**
OCC: Odds of Correct Classification.
(DOCX)

**S6 Table. The associations of EOS count trajectories with ICU mortality and severe AKI incidence in AMI patients.** Model1: unadjusted; Model2: adjusted for age, gender, BMI; Model3: adjusted for age, gender, BMI, SBP, HR, HB, WBC, PLT, Scr, Bun, cTnT, HF, AF, CKD, APSIII, ACEI/ARB, Beta blocker, Antiplatelet drugs, Statin, PCI, CABG.
(DOCX)

**S7 Table. Mediation effect of severe AKI on the association between the EOS count trajectories and 28-day mortality.** Model1: unadjusted; Model2: adjusted for age, gender, BMI; Model3: adjusted for age, gender, BMI, SBP, HR, HB, WBC, PLT, Scr, Bun, cTnT, HF, AF, CKD, APSIII, ACEI/ARB, Beta blocker, Antiplatelet drugs, Statin, PCI, CABG.
(DOCX)

**S8 Table. The associations of EOS% trajectories with 28-day and 1-year mortality in AMI patients.** Model1: unadjusted; Model2: adjusted for age, gender, BMI; Model3: adjusted for age, gender, BMI, SBP, HR, HB, WBC, PLT, Scr, Bun, cTnT, HF, AF, CKD, APSIII, ACEI/ARB, Beta blocker, Antiplatelet drugs, Statin, PCI, CABG.
(DOCX)

**S1 Fig. Restricted cubic spline curves for the association between EOS count and mortality.** (A) Baseline EOS count and 28-day mortality, (B) Baseline EOS count and 1-year mortality, (C) Last EOS count and 28-day mortality and (D) Last EOS count and 1-year mortality.
(JPG)

**S2 Fig. Kaplan–Meier curves according to EOS% trajectory.** (A) 28-day mortality, (B) 1-year mortality.
(JPG)

## Author contributions

**Data curation:** Wen-Liang Shuai, Jia-Hui Huo.

**Formal analysis:** Xiao-Qing Huang.

**Funding acquisition:** Wen-Chao Zhang, Jin-Quan Dai, Jun-Jie Chen, Xiao-xue Xia.

**Investigation:** Jia-Hui Huo, Xiao-xue Xia.

**Methodology:** Xiao-Qing Huang, Jin-Quan Dai, Ming Shen.

**Project administration:** Wen-Chao Zhang, Wen-Liang Shuai, Xiao-Qing Huang, Ming Shen, Zhi-Ming Yang, Xiao-xue Xia.

**Software:** Wen-Chao Zhang, Wen-Liang Shuai.

**Supervision:** Wen-Liang Shuai.

**Validation:** Wen-Chao Zhang.

**Writing – original draft:** Wen-Liang Shuai.

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
