## [Decision Letter · Decision Letter 0]

30 Jan 2026

PONE-D-25-49755Eosinophil Count Trajectories are Associated with the Prognosis of Acute Myocardial Infarction Patients: Insights from ICU Data AnalysisPLOS One

Dear Dr. Xia,

Thank you for submitting your manuscript to PLOS ONE. After careful consideration, we feel that it has merit but does not fully meet PLOS ONE’s publication criteria as it currently stands. Therefore, we invite you to submit a revised version of the manuscript that addresses the points raised during the review process.

This manuscript presents a well-conducted retrospective cohort study using the MIMIC-IV database to investigate the prognostic significance of eosinophil (EOS) count trajectories in critically ill patients with acute myocardial infarction (AMI). By applying group-based trajectory modeling and mediation analysis, the authors provide novel insights into the dynamic role of EOS and its partial mediation through acute kidney injury (AKI). The topic is clinically relevant, methodologically sound, and potentially impactful for ICU risk stratification. However, several issues related to clarity, methodological justification, and presentation should be addressed before the manuscript can be considered for publication.

While the criteria for GBTM model selection are listed (BIC, AIC, AvePP, OCC, subgroup size), the manuscript would benefit from:

-A brief justification for choosing three trajectories over alternative solutions (e.g., 2 or 4 groups).

-Explicit reporting of AvePP and OCC values for each trajectory in the main text or Table 1 (currently relegated to supplementary tables).

The use of random forest imputation is appropriate, but additional details are needed:

-Specify which variables were imputed and their missingness percentages.

-Clarify whether imputation was performed before or after trajectory classification.

-A brief sensitivity analysis excluding imputed variables would strengthen robustness.

-The mediation proportion (≈12%) is modest. This should be emphasized more clearly to avoid overstating the mechanistic role of AKI.

-The temporal relationship between EOS trajectories and AKI onset should be clarified to support the mediation framework.

- The cohort represents critically ill AMI patients requiring ICU care, which limits applicability to general AMI populations  This limitation should be more explicitly acknowledged in the Discussion and Conclusions

The manuscript addresses an important clinical question and employs advanced analytical techniques with largely appropriate methodology. Addressing the points above—particularly regarding trajectory justification, mediation interpretation, and clarity of methods—would substantially strengthen the manuscript and its clinical impact.

We look forward to receiving your revised manuscript.

Kind regards,

Ramada Rateb Khasawneh

Academic Editor

PLOS One

**Journal Requirements:**

1. When submitting your revision, we need you to address these additional requirements. Please ensure that your manuscript meets PLOS ONE's style requirements, including those for file naming. The PLOS ONE style templates can be found at https://journals.plos.org/plosone/s/file?id=wjVg/PLOSOne_formatting_sample_main_body.pdf and https://journals.plos.org/plosone/s/file?id=ba62/PLOSOne_formatting_sample_title_authors_affiliations.pdf 2. Please note that PLOS One has specific guidelines on code sharing for submissions in which author-generated code underpins the findings in the manuscript. In these cases, we expect all author-generated code to be made available without restrictions upon publication of the work. Please review our guidelines at https://journals.plos.org/plosone/s/materials-and-software-sharing#loc-sharing-code and ensure that your code is shared in a way that follows best practice and facilitates reproducibility and reuse. 3. Thank you for stating in your Funding Statement: The project was supported by the Zhejiang Medical Association Special Fund for Clinical Medical Research - Key Project (Grant No. 2023ZYC-Z38).  Please provide an amended statement that declares *all* the funding or sources of support (whether external or internal to your organization) received during this study, as detailed online in our guide for authors at http://journals.plos.org/plosone/s/submit-now. Please also include the statement “There was no additional external funding received for this study.” in your updated Funding Statement. Please include your amended Funding Statement within your cover letter. We will change the online submission form on your behalf. 4. Thank you for stating the following financial disclosure: The project was supported by the Zhejiang Medical Association Special Fund for Clinical Medical Research - Key Project (Grant No. 2023ZYC-Z38).   Please state what role the funders took in the study.  If the funders had no role, please state: "The funders had no role in study design, data collection and analysis, decision to publish, or preparation of the manuscript." If this statement is not correct you must amend it as needed. Please include this amended Role of Funder statement in your cover letter; we will change the online submission form on your behalf. 5. Thank you for uploading your study's underlying data set. Unfortunately, the repository you have noted in your Data Availability statement does not qualify as an acceptable data repository according to PLOS's standards. At this time, please upload the minimal data set necessary to replicate your study's findings to a stable, public repository (such as figshare or Dryad) and provide us with the relevant URLs, DOIs, or accession numbers that may be used to access these data. For a list of recommended repositories and additional information on PLOS standards for data deposition, please see https://journals.plos.org/plosone/s/recommended-repositories. 6. PLOS requires an ORCID iD for the corresponding author in Editorial Manager on papers submitted after December 6th, 2016. Please ensure that you have an ORCID iD and that it is validated in Editorial Manager. To do this, go to ‘Update my Information’ (in the upper left-hand corner of the main menu), and click on the Fetch/Validate link next to the ORCID field. This will take you to the ORCID site and allow you to create a new iD or authenticate a pre-existing iD in Editorial Manager. 7. Your ethics statement should only appear in the Methods section of your manuscript. If your ethics statement is written in any section besides the Methods, please move it to the Methods section and delete it from any other section. Please ensure that your ethics statement is included in your manuscript, as the ethics statement entered into the online submission form will not be published alongside your manuscript. 8. Please upload a new copy of Figures 3, 4, S1 and S2, as the detail is not clear. Please follow the link for more information:  https://journals.plos.org/plosone/s/figures 9. Please include your tables as part of your main manuscript and remove the individual files. Please note that supplementary tables (should remain/ be uploaded) as separate "supporting information" files. 10. Please include captions for your Supporting Information files at the end of your manuscript, and update any in-text citations to match accordingly. Please see our Supporting Information guidelines for more information: http://journals.plos.org/plosone/s/supporting-information. 11. If the reviewer comments include a recommendation to cite specific previously published works, please review and evaluate these publications to determine whether they are relevant and should be cited. There is no requirement to cite these works unless the editor has indicated otherwise.

**Additional Editor Comments:**

This manuscript presents a well-conducted retrospective cohort study using the MIMIC-IV database to investigate the prognostic significance of eosinophil (EOS) count trajectories in critically ill patients with acute myocardial infarction (AMI). By applying group-based trajectory modeling and mediation analysis, the authors provide novel insights into the dynamic role of EOS and its partial mediation through acute kidney injury (AKI). The topic is clinically relevant, methodologically sound, and potentially impactful for ICU risk stratification. However, several issues related to clarity, methodological justification, and presentation should be addressed before the manuscript can be considered for publication.

While the criteria for GBTM model selection are listed (BIC, AIC, AvePP, OCC, subgroup size), the manuscript would benefit from:

-A brief justification for choosing three trajectories over alternative solutions (e.g., 2 or 4 groups).

-Explicit reporting of AvePP and OCC values for each trajectory in the main text or Table 1 (currently relegated to supplementary tables).

The use of random forest imputation is appropriate, but additional details are needed:

-Specify which variables were imputed and their missingness percentages.

-Clarify whether imputation was performed before or after trajectory classification.

-A brief sensitivity analysis excluding imputed variables would strengthen robustness.

-The mediation proportion (≈12%) is modest. This should be emphasized more clearly to avoid overstating the mechanistic role of AKI.

-The temporal relationship between EOS trajectories and AKI onset should be clarified to support the mediation framework.

- The cohort represents critically ill AMI patients requiring ICU care, which limits applicability to general AMI populations  This limitation should be more explicitly acknowledged in the Discussion and Conclusions

The manuscript addresses an important clinical question and employs advanced analytical techniques with largely appropriate methodology. Addressing the points above—particularly regarding trajectory justification, mediation interpretation, and clarity of methods—would substantially strengthen the manuscript and its clinical impact.

Reviewers' comments:

Reviewer's Responses to Questions

**Comments to the Author**

1. Is the manuscript technically sound, and do the data support the conclusions?

Reviewer #1: Yes

Reviewer #2: Yes

Reviewer #3: Yes

2. Has the statistical analysis been performed appropriately and rigorously? 

Reviewer #1: Yes

Reviewer #2: Yes

Reviewer #3: Yes

3. Have the authors made all data underlying the findings in their manuscript fully available?

Reviewer #1: Yes

Reviewer #2: Yes

Reviewer #3: Yes

4. Is the manuscript presented in an intelligible fashion and written in standard English?

Reviewer #1: Yes

Reviewer #2: Yes

Reviewer #3: Yes

5. Review Comments to the Author

**Reviewer #1:** Thank you for giving me the opportunity to review this article. This is a well-conducted retrospective cohort study examining eosinophil trajectories in ICU patients with acute myocardial infarction. The use of group-based trajectory modeling is innovative and the findings are clinically relevant. However, there are several areas requiring clarification and improvement before publication:

1.While the results are compelling, parts of the manuscript imply causal interpretation (e.g., eosinophils “protect” myocardium). Given the retrospective design, trajectory grouping, and biologic complexity, EOS levels may represent disease severity markers rather than mechanistic drivers.

2. The authors appropriately exclude steroid users and eosinophil-driven illnesses. However, subclinical immune conditions or intermittent outpatient steroids may not be captured.

3. The exclusion of patients with ICU stay <24 hours (n=1,056, ~15% of cohort) and patients with <2 EOS measurements within 7 days of ICU admission (n=3,787, >50% of eligible patients) needs justification. What was the rationale behind excluding these cohorts?

4. Although adjustment was comprehensive (as shown in Table 2 and Table S2), patients in the lowest EOS group clearly had worse baseline health. This raises concern that EOS may be a marker of underlying illness severity and residual confounding likely persists.

5. Furthermore, patients in Trajectories 2 and 3 received far more guideline-directed medical therapy (GDMT) and also underwent greater no. of advanced procedures, such as CABG. The survival benefit noted in Trajectory 2 & 3 might be due to superior treatment rather than EOS biology as also shown in Table S2: Beta-blockers (HR 0.21), Statins (HR 0.36), CABG (HR 0.21) are MUCH stronger predictors than EOS trajectories.

6. While the statistical methodology is sound, the clinical interpretation needs strengthening: Trajectory 2 vs. Trajectory 3: Both show rising EOS, yet Trajectory 2 loses significance for long-term mortality after adjustment. What's the biological/clinical explanation? Is there a threshold EOS level needed for sustained cardioprotection?

7. The finding that AKI mediates ~12% of mortality risk is interesting but relies on strong modeling assumptions. Mediation does not prove biological causation and may be influenced by unmeasured confounding between AKI and mortality. Furthermore, the temporal sequence isn't clear -- You define severe AKI as KDIGO stage ≥2 occurring anytime during hospitalization But EOS trajectories evolve over 7 days. When does AKI typically occur relative to EOS trajectory establishment? Is it certain that EOS changes precede AKI, or could AKI itself affect EOS counts?

8. The three EOS trajectories are clearly defined, but the clinical meaning of “low-level steady rise” vs. “medium-level rapid rise” is not fully explored.

9. The authors used random forest imputation for variables with >15% missingness (e.g., BMI, cTnT). Which specific variables were imputed? Was missingness related to outcome (missing not at random)? How was imputation model validated?

10. The authors criticize static scores like SOFA and APACHE II but do not compare the predictive performance of EOS trajectories against these tools. Furthermore, when during the ICU stay can you reliably classify a patient's trajectory? You use 7 days of data to create trajectories But clinicians need risk stratification much earlier as by day 7, many critical decisions have already been made, in which case static scores serve the function by calculation at any point of the admission.

**Reviewer #2:** The authors have presented a very detailed and elaborate report, investigating the link between eosinophil count during ICU admission on mortality and the incidence of acute kidney injury in patients suffering from myocardial infarction. The lone suggestion I have here, is if the authors could describe the connection between AMI and AKI in the abstract as well as the introduction?

**Reviewer #3:** Thank you for inviting me to review this manuscript. This paper addresses an important topic by examining EOS trajectories among ICU patients suffering from AMI. However, several issues require clarification and strengthening.

• Overall, the introduction and the method sections are currently somewhat brief and could benefit from additional background context.

• The data definitions may be clearer if presented in a table rather than solely in narrative text.

• The author mentioned the definition of the trajectory categories in the discussion; however, the author should explain the terms, such as sustained rise, either gradually or rapidly, with values and citation under the method section.

• The author focused on the pathophysiology and immunological mechanisms in the discussion. But the manuscript would benefit from a more detailed discussion of existing discrepancies in the literature regarding the role of the EOS count in AMI, highlighting how the present findings help clarify or contribute to resolving these inconsistencies, would strengthen the discussion and emphasize the study’s contribution. Potential association between EOS levels and cardiovascular pathophysiology can be included in the introduction section instead of discussion.

• The authors acknowledged the limitation related to the use of a database that lacks cardiac function assessments such as echocardiography. However, this limitation is important and may have implications for residual confounding or mechanistic interpretation of the observed associations. The authors may consider discussing this limitation in greater depth and clarifying how the absence of cardiac functional data might influence their findings and conclusions.

6. PLOS authors have the option to publish the peer review history of their article (what does this mean?). If published, this will include your full peer review and any attached files.

Reviewer #1: No

Reviewer #2: No

Reviewer #3: No

---

## [Author Response · Author response to Decision Letter 1]

17 Apr 2026

For detailed reply comments, please see the "response to reviews" document

---

## [Decision Letter · Decision Letter 1]

6 May 2026

Eosinophil Count Trajectories are Associated with the Prognosis of Acute Myocardial Infarction Patients: Insights from ICU Data Analysis

PONE-D-25-49755R1

Dear Dr. Xia,

We’re pleased to inform you that your manuscript has been judged scientifically suitable for publication and will be formally accepted for publication once it meets all outstanding technical requirements.

Kind regards,

Ramada Rateb Khasawneh

Academic Editor

PLOS One

Additional Editor Comments (optional):

Good Luck

Reviewers' comments:

Reviewer's Responses to Questions

**Comments to the Author**

1. If the authors have adequately addressed your comments raised in a previous round of review and you feel that this manuscript is now acceptable for publication, you may indicate that here to bypass the “Comments to the Author” section, enter your conflict of interest statement in the “Confidential to Editor” section, and submit your "Accept" recommendation.

Reviewer #2: All comments have been addressed

Reviewer #3: All comments have been addressed

2. Is the manuscript technically sound, and do the data support the conclusions?

Reviewer #2: Yes

Reviewer #3: Yes

3. Has the statistical analysis been performed appropriately and rigorously? 

Reviewer #2: Yes

Reviewer #3: Yes

4. Have the authors made all data underlying the findings in their manuscript fully available?

Reviewer #2: Yes

Reviewer #3: Yes

5. Is the manuscript presented in an intelligible fashion and written in standard English?

Reviewer #2: Yes

Reviewer #3: Yes

6. Review Comments to the Author

Reviewer #2: The authors have addressed all the comments, suggestions and recommendations presented in previous reviews.

Reviewer #3: (No Response)

7. PLOS authors have the option to publish the peer review history of their article (what does this mean?). If published, this will include your full peer review and any attached files.

Reviewer #2: No

Reviewer #3: No

---

## [Editor Report · Acceptance letter]

PONE-D-25-49755R1

PLOS One

Dear Dr. Xia,

I'm pleased to inform you that your manuscript has been deemed suitable for publication in PLOS One. Congratulations! Your manuscript is now being handed over to our production team.

Kind regards,

on behalf of

Dr. Ramada Rateb Khasawneh

Academic Editor

PLOS One